# How to Initialize your Network?
# Robust Initialization for WeightNorm & ResNets

**Devansh Arpit**[*][†], **Víctor Campos**[*][‡], **Yoshua Bengio**[§]
[†]Salesforce Research, [‡]Barcelona Supercomputing Center,
[§]Montréal Institute for Learning Algorithms, Université de Montréal, CIFAR Senior Fellow
devansharpit@gmail.com, victor.campos@bsc.es

## Abstract

Residual networks (ResNet) and weight normalization play an important role in various deep learning applications. However, parameter initialization strategies have not been studied previously for weight normalized networks and, in practice, initialization methods designed for un-normalized networks are used as a proxy. Similarly, initialization for ResNets have also been studied for un-normalized networks and often under simplified settings ignoring the shortcut connection. To address these issues, we propose a novel parameter initialization strategy that avoids explosion/vanishment of information across layers for weight normalized networks with and without residual connections. The proposed strategy is based on a theoretical analysis using mean field approximation. We run over 2,500 experiments and evaluate our proposal on image datasets showing that the proposed initialization outperforms existing initialization methods in terms of generalization performance, robustness to hyper-parameter values and variance between seeds, especially when networks get deeper in which case existing methods fail to even start training. Finally, we show that using our initialization in conjunction with learning rate warmup is able to reduce the gap between the performance of weight normalized and batch normalized networks.

## 1 Introduction

Parameter initialization is an important aspect of deep network optimization and plays a crucial role in determining the quality of the final model. In order for deep networks to learn successfully using gradient descent based methods, information must flow smoothly in both forward and backward directions [5, 10, 8, 7]. Too large or too small parameter scale leads to information exploding or vanishing across hidden layers in both directions. This could lead to loss being stuck at initialization or quickly diverging at the beginning of training. Beyond these characteristics near the point of initialization itself, we argue that the choice of initialization also has an impact on the final generalization performance. This non-trivial relationship between initialization and final performance emerges because *good* initializations allow the use of *larger* learning rates which have been shown in existing literature to correlate with better generalization [12, 26, 27].

Weight normalization [23] accelerates convergence of stochastic gradient descent optimization by re-parameterizing weight vectors in neural networks. However, previous works have not studied initialization strategies for weight normalization and it is a common practice to use initialization schemes designed for un-normalized networks as a proxy. We study initialization conditions for weight normalized ReLU networks, and propose a new initialization strategy for both plain and residual architectures.

---

[*]Equal contribution. Work done while Víctor Campos was an intern at Salesforce Research.

The main contribution of this work is the theoretical derivation of a novel initialization strategy for weight normalized ReLU networks, with and without residual connections, that prevents information flow from exploding/vanishing in forward and backward pass. Extensive experimental evaluation shows that the proposed initialization increases robustness to network depth, choice of hyper-parameters and seed. When combining the proposed initialization with learning rate warmup, we are able to use learning rates as large as the ones used with batch normalization [11] and significantly reduce the generalization gap between weight and batch normalized networks reported in the literature [4, 25]. Further analysis reveals that our proposal initializes networks in regions of the parameter space that have low curvature, thus allowing the use of large learning rates which are known to correlate with better generalization [12, 26, 27].

## 2 Background and Existing Work

**Weight Normalization:** previous works have considered re-parameterizations that normalize weights in neural networks as means to accelerate convergence. In Arpit et al. [1], the pre- and post-activations are scaled/summed with constants depending on the activation function, ensuring that the hidden activations have 0 mean and unit variance, especially at initialization. However, their work makes assumptions on the distribution of input and pre-activations of the hidden layers in order to make these guarantees. Weight normalization [23] is a simpler alternative, and the authors propose to use a data-dependent initialization [19, 15] for the introduced re-parameterization. This operation improves the flow of information, but its dependence on statistics computed from a batch of data may make it sensitive to the samples used to estimate the initial values.

**Residual Network Architecture:** residual networks (ResNets) [10] have become a cornerstone of deep learning due to their state-of-the-art performance in various applications. However, when using residual networks with weight normalization instead of batch normalization [11], they have been shown to have significantly worse generalization performance. For instance, Gitman and Ginsburg [4] and Shang et al. [25] have shown that ResNets with weight normalization suffer from severe over-fitting and have concluded that batch normalization has an implicit regularization effect.

**Initialization strategies:** there exists extensive literature studying initialization schemes for un-normalized plain networks (c.f. Glorot and Bengio [5], He et al. [9], Saxe et al. [24], Poole et al. [22], Pennington et al. [20, 21], to name some of the most prominent ones). Similarly, previous works have studied initialization strategies for un-normalized ResNets [8, 28, 29], but they lack large scale experiments demonstrating the effectiveness of the proposed approaches and consider a *simplified ResNet* setup where shortcut connections are ignored, even though they play an important role [13]. Zhang et al. [33] propose an initialization scheme for un-normalized ResNets which involves initializing the different types of layers individually using carefully designed schemes. They provide large scale experiments on various datasets, and show that the generalization gap between batch normalized ResNets and un-normalized ResNets can be reduced when using their initialization along with additional domain-specific regularization techniques like cutout [3] and mixup [32]. All the aforementioned works consider un-normalized networks and, to the best of our knowledge, there has been no formal analysis of initialization strategies for weight normalized networks that allow a smooth flow of information in the forward and backward pass.

## 3 Weight Normalized ReLU Networks

We derive initialization schemes for weight normalized networks in the asymptotic setting where network width tends to infinity, similarly to previous analysis for un-normalized networks [5, 10]. We define an $L$ layer weight normalized ReLU network $f_\theta(\mathbf{x}) = \mathbf{h}^L$ recursively, where the $l^{th}$ hidden layer's activation is given by,

$$\mathbf{h}^l := ReLU(\mathbf{a}^l)$$
$$\mathbf{a}^l := \mathbf{g}^l \odot \hat{\mathbf{W}}^l \mathbf{h}^{l-1} + \mathbf{b}^l \quad l \in \{1, 2, \cdots L\} \tag{1}$$

where $\mathbf{a}^l$ are the pre-activations, $\mathbf{h}^l \in \mathbb{R}^{n_l}$ are the hidden activations, $\mathbf{h}^o = \mathbf{x}$ is the input to the network, $\mathbf{W}^l \in \mathbb{R}^{n_l \times n_{l-1}}$ are the weight matrices, $\mathbf{b} \in \mathbb{R}^{n_l}$ are the bias vectors, and $\mathbf{g}^l \in \mathbb{R}^{n_l}$ is a scale factor. We denote the set of all learnable parameters as $\theta = \{(\mathbf{W}^l, \mathbf{g}^l, \mathbf{b}^l)\}_{l=1}^L$. Notation $\hat{\mathbf{W}}^l$

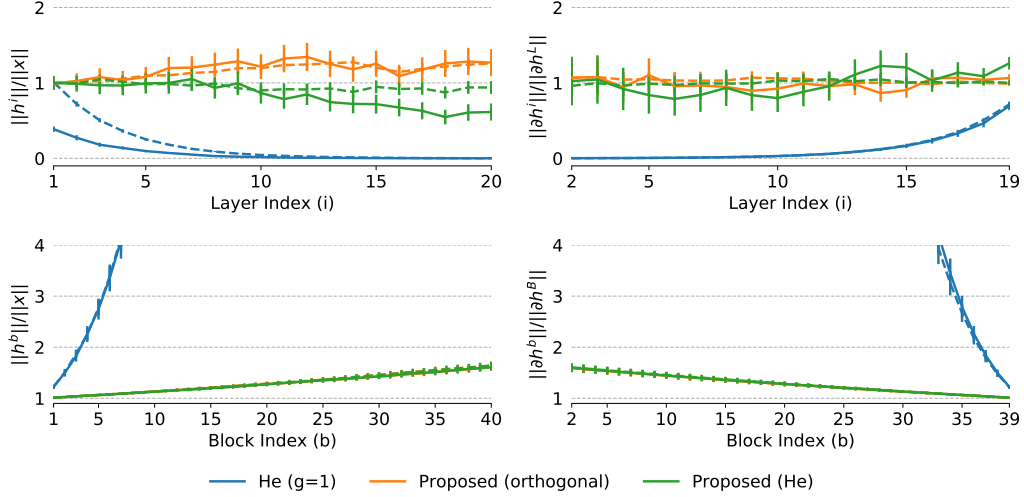

Figure 1: Experiments on weight normalized networks using synthetic data to confirm theoretical predictions. **Top:** feed forward networks. **Bottom:** residual networks. We report results for networks of width $\sim \mathcal{U}(150, 250)$ (solid lines) and width $\sim \mathcal{U}(950, 1050)$ (dashed lines). The proposed initialization prevents explosion/vanishing of the norm of hidden activations (left) and gradients (right) across layers at initialization. For ResNets, norm growth is $\mathcal{O}(1)$ for an arbitrary depth network. Naively initializing $\mathbf{g} = \mathbf{1}$ results in vanishing/exploding signals.

implies that each row vector of $\hat{\mathbf{W}}^l$ has unit norm, i.e.,

$$\hat{\mathbf{W}}_i^l = \frac{\mathbf{W}_i^l}{\|\mathbf{W}_i^l\|_2} \quad \forall i \tag{2}$$

thus $\mathbf{g}_i^l$ controls the norm of each weight vector, whereas $\hat{\mathbf{W}}_i^l$ controls its direction. Finally, we will make use of the notion $\ell(f_\theta(\mathbf{x}), \mathbf{y})$ to represent a differentiable loss function over the output of the network.

**Forward pass:** we first study the forward pass and derive an initialization scheme such that for any given input, the norm of hidden activation of any layer and input norm are asymptotically equal. Failure to do so prevents training to begin, as studied by Hanin and Rolnick [8] for vanilla deep feedforward networks. The theorem below shows that a normalized linear transformation followed by ReLU non-linearity is a norm preserving transform in expectation when proper scaling is used.

**Theorem 1** *Let* $\mathbf{v} = ReLU\left(\sqrt{2n/m} \cdot \hat{\mathbf{R}}\mathbf{u}\right)$, *where* $\mathbf{u} \in \mathbb{R}^n$ *and* $\hat{\mathbf{R}} \in \mathbb{R}^{m \times n}$. *If* $\mathbf{R}_i \overset{i.i.d.}{\sim} P$ *where* $P$ *is any isotropic distribution in* $\mathbb{R}^n$, *or alternatively* $\hat{\mathbf{R}}$ *is a randomly generated matrix with orthogonal rows, then for any fixed vector* $\mathbf{u}$, $\mathbb{E}[\|\mathbf{v}\|^2] = K_n \cdot \|\mathbf{u}\|^2$ *where,*

$$K_n = \begin{cases} \frac{2S_{n-1}}{S_n} \cdot \left(\frac{2}{3} \cdot \frac{4}{5} \cdots \frac{n-2}{n-1}\right) & \text{if } n \text{ is even} \\ \frac{2S_{n-1}}{S_n} \cdot \left(\frac{1}{2} \cdot \frac{3}{4} \cdots \frac{n-2}{n-1}\right) \cdot \frac{\pi}{2} & \text{otherwise} \end{cases} \tag{3}$$

*and* $S_n$ *is the surface area of a unit* $n$-*dimensional sphere.*

The constant $K_n$ seems hard to evaluate analytically, but remarkably, we empirically find that $K_n = 1$ for all integers $n > 1$. Thus applying the above theorem to Eq. 1 implies that every hidden layer in a weight normalized ReLU network is norm preserving for an infinitely wide network if the elements of $\mathbf{g}^l$ are initialized with $\sqrt{2n_{l-1}/n_l}$. Therefore, we can recursively apply the above argument to each layer in a normalized deep ReLU network starting from the input to the last layer and have that the network output norm is approximately equal to the input norm, i.e. $\|f_\theta(\mathbf{x})\| \approx \|\mathbf{x}\|$. Figure 1 (top left) shows a synthetic experiment with a 20 layer weight normalized MLP that empirically confirms the above theory. Details for this experiment can be found in the supplementary material.

**Backward pass:** the goal of studying the backward pass is to derive conditions for which gradients do not explode nor vanish, which is essential for gradient descent based training. Therefore, we

are interested in the value of $\|\frac{\partial \ell(f_\theta(\mathbf{x}), \mathbf{y})}{\partial \mathbf{a}^l}\|$ for different layers, which are indexed by $l$. To prevent exploding/vanishing gradients, the value of this term should be similar for all layers. We begin by writing the recursive relation between the value of this derivative for consecutive layers,

$$\frac{\partial \ell(f_\theta(\mathbf{x}), \mathbf{y})}{\partial \mathbf{a}^l} = \frac{\partial \mathbf{a}^{l+1}}{\partial \mathbf{a}^l} \cdot \frac{\partial \ell(f_\theta(\mathbf{x}), \mathbf{y})}{\partial \mathbf{a}^{l+1}} \tag{4}$$

$$= \mathbf{g}^{l+1} \odot \mathbb{1}(\mathbf{a}^l) \odot \left( \hat{\mathbf{W}}^{l+1^T} \frac{\partial \ell(f_\theta(\mathbf{x}), \mathbf{y})}{\partial \mathbf{a}^{l+1}} \right) \tag{5}$$

We note that conditioned on a fixed $\mathbf{h}^{l-1}$, each dimension of $\mathbb{1}(\mathbf{a}^l)$ in the above equation follows an i.i.d. sampling from Bernoulli distribution with probability 0.5 at initialization. This is formalized in Lemma 1 in the supplementary material. We now consider the following theorem,

**Theorem 2** *Let* $\mathbf{v} = \sqrt{2} \cdot \mathbf{z} \odot \left( \hat{\mathbf{R}}^T \mathbf{u} \right)$, *where* $\mathbf{u} \in \mathbb{R}^m$, $\mathbf{R} \in \mathbb{R}^{m \times n}$ *and* $\mathbf{z} \in \mathbb{R}^n$. *If each* $\mathbf{R}_i \overset{i.i.d.}{\sim} P$ *where* $P$ *is any isotropic distribution in* $\mathbb{R}^n$ *or alternatively* $\hat{\mathbf{R}}$ *is a randomly generated matrix with orthogonal rows and* $\mathbf{z}_i \overset{i.i.d.}{\sim}$ *Bernoulli*(0.5), *then for any fixed vector* $\mathbf{u}$, $\mathbb{E}[\|\mathbf{v}\|^2] = \|\mathbf{u}\|^2$.

In order to apply the above theorem to Eq. 5, we assume that $\mathbf{u} := \frac{\partial \ell(f_\theta(\mathbf{x}), \mathbf{y})}{\partial \mathbf{a}^{l+1}}$ is independent of the other terms, similar to He et al. [10]. This simplifies the analysis by allowing us to treat $\frac{\partial \ell(f_\theta(\mathbf{x}), \mathbf{y})}{\partial \mathbf{a}^{l+1}}$ as fixed and take expectation w.r.t. the other terms, over $\mathbf{W}^l$ and $\mathbf{W}^{l+1}$. Thus $\|\frac{\partial \ell(f_\theta(\mathbf{x}), \mathbf{y})}{\partial \mathbf{a}^l}\| = \|\frac{\partial \ell(f_\theta(\mathbf{x}), \mathbf{y})}{\partial \mathbf{a}^{l+1}}\| \ \forall l$ if we initialize $\mathbf{g}^l = \sqrt{2} \cdot \mathbf{1}$. This also shows that $\frac{\partial \mathbf{a}^{l+1}}{\partial \mathbf{a}^l}$ is a norm preserving transform. Hence applying this theorem recursively to Eq. 5 for all $l$ yields that $\|\frac{\partial \ell(f_\theta(\mathbf{x}), \mathbf{y})}{\partial \mathbf{a}^l}\| \approx \frac{\partial \ell(f_\theta(\mathbf{x}), \mathbf{y})}{\partial \mathbf{a}_L}$ $\forall l$ thereby avoiding gradient explosion/vanishment. Note that the above result is strictly better for orthogonal weight matrices compared with other isotropic distributions (see proof). Figure 1 (top right) shows a synthetic experiment with a 20 layer weight normalized MLP to confirm the above theory. The details for this experiment are provided in the supplementary material.

We also point out that the $\sqrt{2}$ factor that appears in theorems 1 and 2 is due to the presence of ReLU activation. In the absence of ReLU, this factor should be 1. We will use this fact in the next section with the ResNet architecture.

**Implementation details:** since there is a discrepancy between the initialization required by the forward and backward pass, we tested both (and combinations of them) in our preliminary experiments and found the one proposed for the forward pass to be superior. We therefore propose to initialize weight matrices $\mathbf{W}^l$ to be orthogonal[2], $\mathbf{b}^l = \mathbf{0}$, and $\mathbf{g}^l = \sqrt{2n_{l-1}/n_l} \cdot \mathbf{1}$, where $n_{l-1}$ and $n_l$ represent the fan-in and fan-out of the $l^{th}$ layer respectively. Our results apply to both fully-connected and convolutional[3] networks.

## 4  Residual Networks

Similar to the previous section, we derive an initialization strategy for ResNets in the infinite width setting. We define a residual network $\mathcal{R}(\{F_b(\cdot)\}_{b=0}^{B-1}, \theta, \alpha)$ with $B$ residual blocks and parameters $\theta$ whose output is denoted as $f_\theta(\cdot) = \mathbf{h}^B$, and the hidden states are defined recursively as,

$$\mathbf{h}^{b+1} := \mathbf{h}^b + \alpha F_b(\mathbf{h}^b) \quad b \in \{0, 1, \dots, B-1\} \tag{6}$$

where $\mathbf{h}^0 = \mathbf{x}$ is the input, $\mathbf{h}^b$ denotes the hidden representation after applying $b$ residual blocks and $\alpha$ is a scalar that scales the output of the $b$-th residual blocks. The $b$-th $\in \{0, 1, \dots, B-1\}$ residual block $F_b(\cdot)$ is a feed-forward ReLU network. We discuss how to deal with shortcut connections during initialization separately. We use the notation $<\cdot, \cdot>$ to denote dot product between the argument vectors.

**Forward pass**: here we derive an initialization strategy for residual networks that prevents information in the forward pass from exploding/vanishing independent of the number of residual blocks, assuming that each residual block is initialized such that it preserves information in the forward pass.

**Theorem 3** *Let $\mathcal{R}(\{F_b(\cdot)\}_{b=0}^{B-1}, \theta, \alpha)$ be a residual network with output $f_\theta(\cdot)$. Assume that each residual block $F_b(.)$ ($\forall b$) is designed such that at initialization, $\|F_b(\mathbf{h})\| = \|\mathbf{h}\|$ for any input $\mathbf{h}$ to the block, and $<\mathbf{u}, F_b(\mathbf{u})> \approx 0$. If we set $\alpha = 1/\sqrt{B}$, then $\exists c \in [\sqrt{2}, \sqrt{e}]$, such that,*

$$\|f_\theta(\mathbf{x})\| \approx c \cdot \|\mathbf{x}\| \tag{7}$$

The assumption $<\mathbf{u}, F_b(\mathbf{u})> \approx 0$ is reasonable because at initialization, $F_b(\mathbf{u})$ is a random transformation in a high dimensional space which will likely rotate a vector to be orthogonal to itself. To understand the rationale behind the second assumption, $\|F_b(\mathbf{h})\| = \|\mathbf{h}\|$, recall that $F_b(.)$ is essentially a non-residual network. Therefore we can initialize each such block using the scheme developed in Section 3 which due to Theorem 1 (see discussion below it) will guarantee that the norm of the output of $F_b(\cdot)$ equals the norm of the input to the block. Figure 1 (bottom left) shows a synthetic experiment with a 40 block weight normalized ResNet to confirm the above theory. The ratio of norms of output to input lies in $[\sqrt{2}, \sqrt{e}]$ independent of the number of residual blocks exactly as predicted by the theory. The details for this experiment can be found in the supplementary material.

**Backward pass:** we now study the backward pass for residual networks.

**Theorem 4** *Let $\mathcal{R}(\{F_b(\cdot)\}_{b=0}^{B-1}, \theta, \alpha)$ be a residual network with output $f_\theta(\cdot)$. Assume that each residual block $F_b(\cdot)$ ($\forall b$) is designed such that at initialization, $\|\frac{\partial F_b(\mathbf{h}^b)}{\partial \mathbf{h}^b}\mathbf{u}\| = \|\mathbf{u}\|$ for any fixed input $\mathbf{u}$ of appropriate dimensions, and $<\frac{\partial \ell}{\partial \mathbf{h}^b}, \frac{\partial \mathbf{F}_{b-1}}{\partial \mathbf{h}^{b-1}} \cdot \frac{\partial \ell}{\partial \mathbf{h}_b}> \approx 0$. If $\alpha = \frac{1}{\sqrt{B}}$, then $\exists c \in [\sqrt{2}, \sqrt{e}]$, such that,*

$$\|\frac{\partial \ell}{\partial \mathbf{h}^1}\| \approx c \cdot \|\frac{\partial \ell}{\partial \mathbf{h}^B}\| \tag{8}$$

The above theorem shows that scaling the output of the residual block with $1/\sqrt{B}$ prevents explosion/vanishing of gradients irrespective of the number of residual blocks. The rationale behind the assumptions is similar to that given for the forward pass above. Figure 1 (bottom right) shows a synthetic experiment with a 40 block weight normalized ResNet to confirm the above theory. Once again, the ratio of norms of gradient w.r.t. input to output lies in $[\sqrt{2}, \sqrt{e}]$ independent of the number of residual blocks exactly as predicted by the theory. The details can be found in the supplementary material.

**Shortcut connections:** a ResNet often has $K$ *stages* [10], where each stage is characterized by one *shortcut* connection and $B_k$ residual blocks, leading to a total of $\sum_{k=1}^{K} B_k$ blocks. In order to account for shortcut connections, we need to ensure that the input and output of each *stage* in a ResNet are at the same scale; the same argument applies during the backward pass. To achieve this, we scale the output of the residual blocks in each stage using the total number of residual blocks in that stage. Then theorems 3 and 4 treat each stage of the network as a ResNet and normalize the flow of information in both directions to be independent of the number of residual blocks.

**Implementation details:** we consider ResNets with shortcut connections and architecture design similar to that proposed in [10] with the exception that our residual block structure is Conv→ReLU→Conv, similar to $B(3, 3)$ blocks in [31], as illustrated in the supplementary material[4]. Weights of all layers in the network are initialized to be orthogonal and biases are set to zero. The gain parameter of weight normalization is initialized to be $\mathbf{g} = \sqrt{\gamma \cdot \text{fan-in}/\text{fan-out}} \cdot \mathbf{1}$. We set $\gamma = 1/B_k$ for the last convolutional layer of each residual block in the $k$-th stage[5]. For the rest of layers we follow the strategy derived in Section 3, with $\gamma = 2$ when the layer is followed by ReLU, and $\gamma = 1$ otherwise.

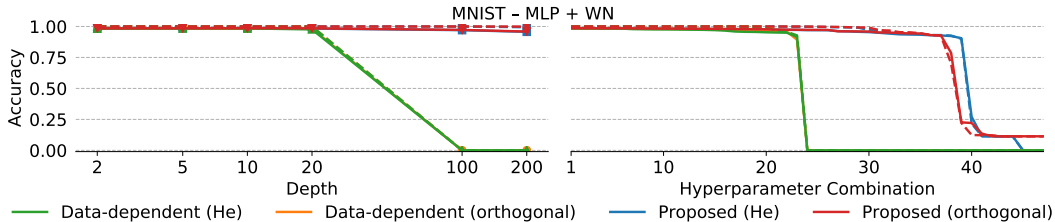

Figure 2: Results for MLPs on MNIST. Dashed lines denote train accuracy, and solid lines denote test accuracy. The accuracy of diverged runs is set to 0. **Left:** Accuracy as a function of depth. A held-out validation set is used to select the best model for each configuration. **Right:** Accuracy for each job in our hyperparameter sweep, depicting robustness to hyperparameter configurations.

## 5 Experiments

We study the impact of initialization on weight normalized networks across a wide variety of configurations. Among others, we compare against the data-dependent initialization proposed by Salimans and Kingma [23], which initializes $\mathbf{g}$ and $\mathbf{b}$ so that all pre-activations in the network have zero mean and unit variance based on estimates collected from a single minibatch of data.

Code is publicly available at `https://github.com/victorcampos7/weightnorm-init`. We refer the reader to the supplementary material for detailed description of the hyperparameter settings for each experiment, as well as for initial reinforcement learning results.

### 5.1 Robustness Analysis of Initialization methods– Depth, Hyper-parameters and Seed

The difficulty of training due to exploding and vanishing gradients increases with network depth. In practice, depth often complicates the search for hyperparameters that enable successful optimization, if any. This section presents a thorough evaluation of the impact of initialization on different network architectures for increasing depths, as well as their robustness to hyperparameter configurations. We benchmark fully-connected networks on MNIST [17], whereas CIFAR-10 [16] is considered for convolutional and residual networks. We tune hyperparameters individually for each network depth and initialization strategy on a set of held-out examples, and report results on the test set. We refer the reader to the supplementary material for a detailed description of the considered hyperparameters.

**Fully-connected networks:** results in Figure 2 (left) show that the data-dependent initialization can be used to train networks of up to depth 20, but training diverges for deeper nets even when using very small learning rates, e.g. $10^{-5}$. On the other hand, we managed to successfully train very deep networks with up to 200 layers using the proposed initialization. When analyzing all runs in the grid search, we observe that the proposed initialization is more robust to the particular choice of hyperparameters (Figure 2, right). In particular, the proposed initialization allows using learning rates up to $10\times$ larger for most depths.

**Convolutional networks:** we adopt a similar architecture to that in Xiao et al. [30], where all layers have $3 \times 3$ kernels and a fixed width. The two first layers use a stride of 2 in order to reduce the memory footprint. Results are depicted in Figure 3 (left) and show a similar trend to that observed for fully-connected nets, with the data-dependent initialization failing at optimizing very deep networks.

**Residual networks:** we construct residual networks of varying depths by controlling the number of residual blocks per stage in the wide residual network (WRN) architecture with $k = 1$. Training networks with thousands of layers is computationally intensive, so we measure the test accuracy after a single epoch of training [33]. We consider two additional baselines for these experiments: (1) the default initialization in PyTorch[6], which initializes $g_i = \|\mathbf{W}_i\|_2$, and (2) a modification of the initialization proposed by Hanin and Rolnick [8] to fairly adapt it to weight normalized multi-stage ResNets. For the $k$-th stage with $B_k$ blocks, the stage-wise Hanin scheme initializes the gain of the last convolutional layer in each block as $\mathbf{g} = 0.9^b \mathbf{1}$, where $b \in \{1, \ldots, B_k\}$ refers to the block number within a stage. All other parameters are initialized in a way identical to our proposal, so that information across the layers within residual blocks remains preserved. We report results over 5 random seeds for each configuration in Figure 3 (right), which shows that the proposed

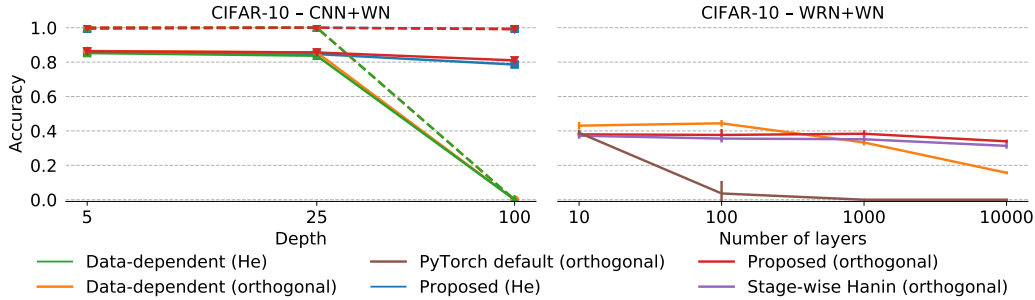

Figure 3: Accuracy as a function of depth on CIFAR-10 for CNNs (**left**), and WRNs (**right**). Dashed lines denote train accuracy, and solid lines denote validation accuracy. Note that WRNs are trained for a single epoch due to the computational burden of training extremely deep networks.

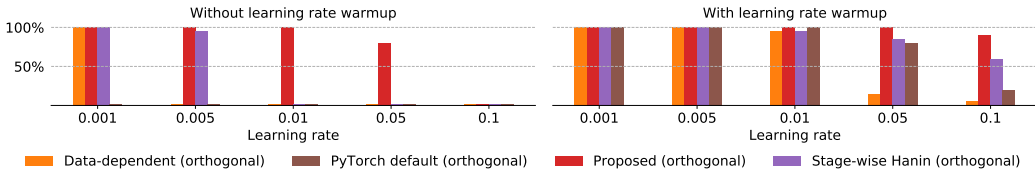

Figure 4: Robustness to seed of different initialization schemes on WRN-40-10. We launch 20 training runs for every configuration, and measure the percentage of runs that reach epoch 3 without diverging. Weight normalized ResNets benefit from learning rate warmup, which enables the usage of higher learning rates. The proposed initialization is the most robust scheme across all configurations.

initialization achieves similar accuracy rates across the wide range of evaluated depths. PyTorch's default initialization diverges for most depths, and the data-dependent baseline converges significantly slower for deeper networks due to the small learning rates used in order to avoid divergence. Despite the stage-wise Hanin strategy and the proposed initialization achieve similar accuracy rates, we were able to use an order of magnitude larger learning rates with the latter, which denotes an increased robustness against hyperparameter configurations.

To further evaluate the robustness of each initialization strategy, we train WRN-40-10 networks for 3 epochs with different learning rates, with and without learning rate warmup [6]. We repeat each experiment 20 times using different random seeds, and report the percentage of runs that successfully completed all 3 epochs without diverging in Figure 4. We observed that learning rate warmup greatly improved the range of learning rates that work well for all initializations, but the proposed strategy manages to train more robustly across all tested configurations.

## 5.2 Comparison with Batch Normalization

Existing literature has pointed towards an implicit regularization effect of batch normalization [18], which prevented weight normalized models from matching the final performance of batch normalized ones [4]. On the other hand, previous works have shown that larger learning rates facilitate finding wider minima which correlate with better generalization performance [14, 12, 26, 27], and the proposed initialization and learning rate warmup have proven very effective in stabilizing training for high learning rates. This section aims at evaluating the final performance of weight normalized networks trained with high learning rates, and compare them with batch normalized networks.

We evaluate models on CIFAR-10 and CIFAR-100. We set aside 10% of the training data for hyperparameter tuning, whereas some previous works use the test set for such purpose [10, 31]. This difference in the experimental setup explains why the achieved error rates are slightly larger than those reported in the literature. For each architecture we use the default hyperparameters reported in literature for batch normalized networks, and tune only the initial learning rate for weight normalized models.

Results in Table 1 show that the proposed initialization scheme, when combined with learning rate warmup, allows weight normalized residual networks to achieve comparable error rates to their batch normalized counterparts. We note that previous works reported a large generalization gap between

weight and batch normalized networks [25, 4]. The only architecture for which the batch normalized variant achieves a superior performance is WRN-40-10, for which the weight normalized version is not able to completely fit the training set before reaching the epoch limit. This phenomena is different to the generalization gap reported in previous works, and might be caused by sub-optimal learning rate schedules that were tailored for networks with batch normalization.

Table 1: Comparison between Weight Normalization with proposed initialization and Batch Normalization. Results are reported as mean and std over 5 runs.

| Dataset | Architecture | Method | Test Error (%) |
|---|---|---|---|
| CIFAR-10 | ResNet-56 | WN w/ datadep init | $9.19 \pm 0.24$ |
| | | WN w/ proposed init | $7.87 \pm 0.14$ |
| | | WN w/ proposed init + warmup | $7.20 \pm 0.12$ |
| | | BN (He et al. [10]) | 6.97 |
| | ResNet-110 | WN w/ datadep init | $9.33 \pm 0.10$ |
| | | WN w/ proposed init | $7.71 \pm 0.14$ |
| | | WN w/ proposed init + warmup | $6.69 \pm 0.11$ |
| | | WN (Shang et al. [25]) | 7.46 |
| | | BN (He et al. [10]) | $6.61 \pm 0.16$ |
| | WRN-40-10 | WN w/ datadep init + cutout | $6.10 \pm 0.23$ |
| | | WN w/ proposed init + cutout | $4.74 \pm 0.14$ |
| | | WN w/ proposed init + cutout + warmup | $4.75 \pm 0.08$ |
| | | BN w/ orthogonal init + cutout | $3.53 \pm 0.38$ |
| CIFAR-100 | ResNet-164 | WN w/ datadep init + cutout | $30.26 \pm 0.51$ |
| | | WN w/ proposed init + cutout | $27.30 \pm 0.49$ |
| | | WN w/ proposed init + cutout + warmup | $25.31 \pm 0.26$ |
| | | BN w/ orthogonal init + cutout | $25.52 \pm 0.17$ |

## 5.3 Initialization Method and Generalization Gap

The motivation behind designing good parameter initialization is mainly for better optimization at the beginning of training, and it is not apparent why our initialization is able to reduce the generalization gap between weight normalized and batch normalized networks [4, 25]. On this note we point out that a number of papers have shown how using stochastic gradient descent (SGD) with larger learning rates facilitate finding wider minima which correlate with better generalization performance [14, 12, 26, 27]. Additionally, it is often not possible to use large learning rates with weight normalization with traditional initializations. Therefore we believe that the use of large learning rate allowed by our initialization played an important role in this aspect. In order to understand why our initialization allows using large learning rates compared with existing ones, we compute the (log) spectral norm of the Hessian at initialization (using Power method) for the various initialization methods considered in our experiments using 10% of the training samples. They are shown in Table 2. We find that the local curvature (spectral norm) is smallest for the proposed initialization. These results are complementary to the seed robustness experiment shown in Figure 4.

Table 2: Log (base 10) spectral norm of Hessian at initialization for different initializations. Smaller values imply lower curvature. *N/A* means that the computation diverged. The proposed strategy initializes at a point with lowest curvature, which explains why larger learning rates can be used.

| Dataset | Model | PyTorch default | Data-dependent | Stage-wise Hanin | Proposed |
|---|---|---|---|---|---|
| CIFAR-10 | WRN-40-10 | $4.68 \pm 0.60$ | $3.01 \pm 0.02$ | $7.14 \pm 0.72$ | $1.31 \pm 0.12$ |
| CIFAR-100 | ResNet-164 | $9.56 \pm 0.54$ | $2.68 \pm 0.09$ | N/A | $1.56 \pm 0.18$ |

## 6 Conclusion and Future Work

Weight normalization (WN) is frequently used in different network architectures due to its simplicity. However, the lack of existing theory on parameter initialization of weight normalized networks has led practitioners to arbitrarily pick existing initializations designed for un-normalized networks. To address this issue, we derived parameter initialization schemes for weight normalized networks, with

and without residual connections, that avoid explosion/vanishment of information in the forward and backward pass. To the best of our knowledge, no prior work has formally studied this setting. Through thorough empirical evaluation, we showed that the proposed initialization increases robustness to network depth, choice of hyper-parameters and seed compared to existing initialization methods that are not designed specifically for weight normalized networks. We found that the proposed scheme initializes networks in low curvature regions, which enable the use of large learning rates. By doing so, we were able to significantly reduce the performance gap between batch and weight normalized networks which had been previously reported in the literature. Therefore, we hope that our proposal replaces the current practice of choosing arbitrary initialization schemes for weight normalized networks.

We believe our proposal can also help in achieving better performance using WN in settings which are not well-suited for batch normalization. One such scenario is training of recurrent networks in backpropagation through time settings, which often suffer from exploding/vanishing gradients, and batch statistics are timestep-dependent [2]. The current analysis was done for feedforward networks, and we plan to extend it to the recurrent setting. Another application where batch normalization often fails is reinforcement learning, as good estimates of activation statistics are not available due to the online nature of some of these algorithms. We confirmed the benefits of our proposal in preliminary reinforcement learning experiments, which can be found in the supplementary material.

## Acknowledgments

We would like to thank Giancarlo Kerg for proofreading the paper. DA was supported by IVADO during his time at Mila where part of this work was done.

## Footnotes

[2]We note that Saxe et al. [24] propose to initialize weights of un-normalized deep ReLU networks to be orthogonal with scale $\sqrt{2}$. Our derivation and proposal is for weight normalized ReLU networks where we study both Gaussian and orthogonal initialization and show the latter is superior.

[3]For convolutional layers with kernel size $k$ and $c$ channels, we define $n_{l-1} = k^2 c_{l-1}$ and $n_l = k^2 c_l$ [9].

[4]More generally, our residual block design principle is $D \times [\text{Conv} \rightarrow \text{ReLU} \rightarrow] \text{Conv}$, where $D \in \mathbb{Z}$.

[5]We therefore absorb $\alpha$ in Eq. 6 into the gain parameter $\mathbf{g}$.

[6]`https://pytorch.org/docs/stable/_modules/torch/nn/utils/weight_norm.html`

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
