[Supplementary Material · supplementary_material.pdf]

# – Supplementary Material –
# How to Initialize your Network?
# Robust Initialization for WeightNorm & ResNets

**Devansh Arpit**[*][†]**, Víctor Campos**[*][‡]**, Yoshua Bengio**[§]
[†]Salesforce Research, [‡]Barcelona Supercomputing Center,
[§]Montréal Institute for Learning Algorithms, Université de Montréal, CIFAR Senior Fellow
devansharpit@gmail.com, victor.campos@bsc.es

[*]Equal contribution. Work done while Víctor Campos was an intern at Salesforce Research.

# A  Experimental setup

## A.1  Details about Figure 1 (top)

We use a weight normalized 20 layer MLP with 1000 randomly generated input samples in $\mathbf{R}^{500}$. We test three initialization strategies. (1) He initialization [3] for the weight matrices and the gain parameter $\mathbf{g}$ for all layers are initialized to 1. (2) Proposed initialization, where weights are initialized to be orthogonal and gains are set as $\sqrt{2n_{l-1}/n_l}$. (3) Proposed initialization, where weights are initialized using He initialization and gains are set as $\sqrt{2n_{l-1}/n_l}$. In all cases biases are set to 0. At initialization itself, we forward propagate the 1000 randomly generated input samples, measure the norm of hidden activations, and compute the mean and standard deviation of the ratio of norm of hidden activation to the norm of the input. This is shown in Figure 1 (top left). In Figure 1 (top right), we similarly record the norm of hidden activation gradient by backpropagating 1000 random error vectors, and measure the ratio of the norm of hidden activation gradient to the norm of the error vector. We find that the proposed initialization preserves norm in both directions while vanilla He initialization fails. This shows the importance of proper initialization of the $\gamma$ parameter of weight normalization.

## A.2  Details about Figure 1 (bottom)

We use a weight normalized ResNet with 40 residual blocks with 1000 randomly generated input samples in $\mathbf{R}^{500}$. The network architecture is exactly as described in Eq. 6, with a residual block composed of two fully connected (FC) layers, i.e. FC1 $\rightarrow$ ReLU $\rightarrow$ FC2. The weight normalization layers are inserted after FC layers. We test three initialization strategies. (1) He initialization [3] for all the weight matrices, and gain parameter $\mathbf{g} = \mathbf{1}$. (2) Proposed initialization where weights are initialized to be orthogonal and gains are set as $\sqrt{2 \cdot \text{fan-in}/\text{fan-out}}$ for FC1 and $\sqrt{\text{fan-in}/(40 \cdot \text{fan-out})}$ for FC2. (3) Proposed initialization where weights are initialized using He initialization and gains are set same as in the previous case. In all cases biases are set to 0. At initialization itself, we forward propagate the 1000 randomly generated input samples, measure the norm of hidden activations $\mathbf{h}^b$ and compute the mean and standard deviation of the ratio of norm of hidden activation to the norm of the input $\mathbf{x}$. This is shown in Figure 1 (bottom left). In Figure 1 (bottom right), we similarly record the norm of hidden activation gradient by backpropagating 1000 random error vectors and measure the ratio of the norm of hidden activation gradient $\frac{\partial \ell}{\partial \mathbf{h}^b}$ to the norm of the error vector $\frac{\partial \ell}{\partial \mathbf{h}^B}$. We find that the proposed initialization preserves norm in both directions while vanilla He initialization fails. This shows the importance of proper initialization of the $\mathbf{g}$ parameter of weight normalization.

Table 1: Hyperparameters for MNIST experiments. Values between brackets were used in the grid search. Learning rate of 0.00001 was considered for depths 100 and 200 only.

| Parameter | Value |
|---|---|
| Data split | 10% of the original train is set aside for validation purposes |
| Number of hidden layers | $\{2, 5, 10, 20, 100, 200\}$ |
| Size of hidden layers | $\{512, 1024\}$ |
| Number of epochs | 150 |
| Initial learning rate | $\{0.1, 0.01, 0.001, 0.0001, 0.00001^*\}$ |
| Learning rate schedule | Decreased by $10\times$ at epochs 50 and 100 |
| Batch size | 128 |
| Weight decay | 0.0001 |
| Optimizer | SGD with momentum $= 0.9$ |

# B  Reinforcement Learning experiments

Despite its tremendous success in supervised learning applications, Batch Normalization [4] is seldom used in reinforcement learning (RL), as the online nature of some of the methods and the strong correlation between consecutive batches hinder its performance. These properties suggest the need

Table 2: Hyperparameters for CNN experiments on CIFAR-10. Values between brackets were used in the grid search. Learning rate of 0.001 was considered for depth 100 only.

| Parameter | Value |
|---|---|
| Data split | 10% of the original train is set aside for validation purposes |
| Architecture | $2 \times [\text{Conv2D } 3 \times 3/2, 512]$ <br> $(N-2) \times [\text{Conv2D } 3 \times 3/1, 512]$ <br> Global Average Pooling <br> 10-d Linear, softmax |
| Number of hidden layers (N) | $\{5, 25, 100\}$ |
| Number of epochs | 500 |
| Initial learning rate | $\{0.01, 0.001^*\}$ |
| Learning rate schedule | Decreased by $10\times$ at epoch 166 |
| Batch size | 100 |
| Weight decay | $\{0.001, 0.0001\}$ |
| Optimizer | SGD without momentum |

| Group name | Output size | Block type |
|---|---|---|
| conv1 | 32×32 | [Conv2D 3×3, 16×k] |
| conv2 | 32×32 | $N \times \begin{bmatrix} \text{Conv2D } 3\times3, 16\times\text{k} \\ \text{ReLU} \\ \text{Conv2D } 3\times3, 16\times\text{k} \end{bmatrix}$ |
| conv3 | 16×16 | $N \times \begin{bmatrix} \text{Conv2D } 3\times3, 32\times\text{k} \\ \text{ReLU} \\ \text{Conv2D } 3\times3, 32\times\text{k} \end{bmatrix}$ |
| conv4 | 8×8 | $N \times \begin{bmatrix} \text{Conv2D } 3\times3, 64\times\text{k} \\ \text{ReLU} \\ \text{Conv2D } 3\times3, 64\times\text{k} \end{bmatrix}$ |
| out | 1×1 | [average pooling, 10-d fc, softmax] |

Figure 1: **Left:** Architecture of Wide Residual Networks considered in this work. Downsampling is performed through strided convolutions by the first layers in groups `conv3` and `conv4`. **Right:** Structure of a residual block. Note that there is no non-linearity after residual conections, unlike [3].

for normalization techniques like Weight Normalization [6], which are able to accelerate and stabilize training of neural networks without relying on minibatch statistics.

We consider the Asynchronous Advantage Actor Critic (A3C) algorithm [5], which maintains a policy and a value function estimate which are updated asynchronously by different workers collecting experience in parallel. Updates are estimated based on $n$-step returns from each worker, resulting in highly correlated batches of $n$ samples, whose impact is mitigated through the asynchronous nature of updates. This setup is not well suited for Batch Normalization and, to the best of our knowledge, no prior work has successfully applied it to this type of algorithm.

We evaluate agents using Atari environments in the Arcade Learning Environment [1]. Our initial experiments with the deep residual architecture introduced by Espeholt et al. [2] show that adding Weight Normalization improves convergence speed and robustness to hyperparameter configurations across different environments. However, we did not observe important differences between initialization schemes for these weight normalized models. Despite being significantly deeper than previous architectures used in RL, this model is still relatively shallow for supervised learning standards, and we observed in our computer vision experiments that performance differences arise for deeper architectures or high learning rates. The latter is known to cause catastrophic performance degradation in deep RL due to excessively large policy updates [7], so we opt for building a much deeper residual network with 100 layers. Collecting experience with such a deep policy is a very slow process even when using GPU workers. Given this computational burden, we use hyperparameters tuned in initial

Table 3: Hyperparameters for WRN experiments on CIFAR-10. Values between brackets were used in the grid search. Learning rates smaller than $10^{-5}$ were considered for $N = 1666$ (10,000 layers) only.

| Parameter | Value |
|---|---|
| Data split | 10% of the original train is set aside for validation purposes |
| WRN's $N$ (residual blocks per stage) | $\{1, 16, 166, 1666\}$ |
| WRN's $k$ (width factor) | 1 |
| Number of epochs | 1 |
| Initial learning rate | $\{7 \cdot 10^{-1}, 3 \cdot 10^{-1}, 1 \cdot 10^{-1}, \ldots, 10^{-5}, \ldots, 10^{-7*}\}$ |
| Batch size | 128 |
| Weight decay | 0.0005 |
| Optimizer | SGD with momentum $= 0.9$ |

experiments for the deep network introduced by Espeholt et al. [2], and report initial results in one of the simplest environments[2] in Figure 2.

Figure 2: Learning progress in Pong. Shading shows maximum and minimum over 3 random seeds, while dark lines indicate the mean. Weight normalization with the proposed initialization improves convergence speed and reduces variance across seeds. These results highlight the importance of initialization in weight normalized networks, as using the default initialization in PyTorch prevents training to start.

We observe that the weight normalized policy with the proposed initialization manages to solve the task much faster than the un-normalized architecture. Perhaps surprisingly, the weight normalized policy with the sub-optimal initialization is not able to solve the environment in the given timestep budget, and it performs even worse than the un-normalized policy. These results highlight the importance of proper initialization even when using normalization techniques.

The deep network architecture considered in this experiment is excessively complex for the considered task, which can be solved with much smaller networks. However, with the development of ever complex environments [9] and distributed learning algorithms that can take advantage of massive computational resources [2], recent results have shown that RL can benefit from techniques that have found success in the supervised learning community, such as deeper residual networks [8, 2]. The aforementioned findings suggest that RL applications could benefit from techniques that help training very deep networks robustly in the future.

## C   Proofs

**Theorem 1** *Let* $\mathbf{v} = ReLU\left(\sqrt{\frac{2n}{m}} \cdot \hat{\mathbf{R}}\mathbf{u}\right)$, *where* $\mathbf{u} \in \mathbb{R}^n$ *and* $\hat{\mathbf{R}} \in \mathbb{R}^{m \times n}$. *If* $\mathbf{R}_i \overset{i.i.d.}{\sim} P$ *where* $P$ *is any isotropic distribution in* $\mathbb{R}^n$, *or alternatively* $\hat{\mathbf{R}}$ *is a randomly generated matrix with orthogonal rows, then for any fixed vector* $\mathbf{u}$, $\mathbb{E}[\|\mathbf{v}\|^2] = K_n \cdot \|\mathbf{u}\|^2$ *where,*

$$K_n = \begin{cases} \frac{2S_{n-1}}{S_n} \cdot \left(\frac{2}{3} \cdot \frac{4}{5} \cdots \frac{n-2}{n-1}\right) & \text{if } n \text{ is odd} \\ \frac{2S_{n-1}}{S_n} \cdot \left(\frac{1}{2} \cdot \frac{3}{4} \cdots \frac{n-2}{n-1}\right) \cdot \frac{\pi}{2} & \text{otherwise} \end{cases} \tag{1}$$

*and* $S_n$ *is the surface are of a unit* $n$-*dimensional sphere.*

**Proof**: *During the proof, take note of the distinction between the notations* $\hat{\mathbf{R}}_i$ *and* $\mathbf{R}_i$. *Our goal is to compute,*

$$\mathbb{E}[\|\mathbf{v}\|^2] = \mathbb{E}[\sum_{i=1}^m v_i^2] \tag{2}$$

$$= \sum_{i=1}^m \mathbb{E}[v_i^2] \tag{3}$$

*Suppose the weights are randomly generated to be orthogonal with uniform probability over all rotations. Due to the linearity of expectation, when taking the expectation of any unit* $v_i$ *over the randomly generated orthogonal weight matrix, the expectation marginalizes over all the rows of the weight matrix except the* $i^{th}$ *row. As a consequence, for each unit* $i$, *the expectation is over an isotropic random variable since the orthogonal matrix is generated randomly with uniform probability over all rotations. Therefore, we can equivalently write,*

$$\mathbb{E}[\|\mathbf{v}\|^2] = m\mathbb{E}[v_i^2] \tag{4}$$

*Note that the above equality would trivially hold if all rows of the weight matrix were sampled i.i.d. from an isotropic distribution. In other words, the above equality holds irrespective of the two choice of distributions used for sampling the weight matrix.*

*We have,*

$$\mathbb{E}[v_i^2] = \mathbb{E}[\max(0, \sqrt{\frac{2n}{m}} \cdot \hat{\mathbf{R}}_i^T \mathbf{u})^2] \tag{5}$$

$$= \int_{\mathbf{R}_i} p(\mathbf{R}_i) \max(0, \sqrt{\frac{2n}{m}} \cdot \|\mathbf{u}\| \cos\theta)^2 \tag{6}$$

*where* $p(\mathbf{R}_i)$ *denotes the probability distribution of the random variable* $\mathbf{R}_i$, *and* $\theta$ *is the angle between vectors* $\hat{\mathbf{R}}_i$ *and* $\mathbf{u}$. *Hence* $\theta$ *is a function of* $\hat{\mathbf{R}}_i$. *Since* $\mathbf{R}_i$ *is sampled from an isotropic distribution, the direction and scale of* $\mathbf{R}_i$ *are independent. Thus,*

$$\int p(\mathbf{R}_i) \max(0, \sqrt{\frac{2n}{m}} \cdot \|\mathbf{u}\| \cos\theta)^2 = \int_{\mathbf{R}_i} p(\|\mathbf{R}_i\|) \int_{\hat{\mathbf{R}}_i} p(\hat{\mathbf{R}}_i) \max(0, \sqrt{\frac{2n}{m}} \cdot \|\mathbf{u}\| \cos\theta)^2 \tag{7}$$

$$= \int_{\hat{\mathbf{R}}_i} p(\hat{\mathbf{R}}_i) \max(0, \sqrt{\frac{2n}{m}} \cdot \|\mathbf{u}\| \cos\theta)^2 \tag{8}$$

$$= \frac{2n}{m} \cdot \|\mathbf{u}\|^2 \int_{\hat{\mathbf{R}}_i} p(\hat{\mathbf{R}}_i) \max(0, \cos\theta)^2 \tag{9}$$

*Since* $P$ *is an isotropic distribution in* $\mathbb{R}^n$, *the likelihood of all directions is uniform. It essentially means that* $p(\hat{\mathbf{R}}_i)$ *can be seen as a uniform distribution over the surface area of a unit* $n$-*dimensional sphere. We can therefore re-parameterize* $p(\hat{\mathbf{R}}_i)$ *in terms of* $\theta$ *by aggregating the density* $p(\hat{\mathbf{R}}_i)$ *over all points on this* $n$-*dimensional sphere at a fixed angle* $\theta$ *from the vector* $\mathbf{u}$. *This is similar to the idea of Lebesgue integral. To achieve this, we note that all the points on the* $n$-*dimensional sphere at a constant angle* $\theta$ *from* $\mathbf{u}$ *lie on an* $(n-1)$-*dimensional sphere of radius* $\sin\theta$. *Thus the aggregate*

*density at an angle $\theta$ from $\mathbf{u}$ is the ratio of the surface area of the $(n-1)$-dimensional sphere of radius $\sin\theta$ and the surface area of the unit $(n)$-dimensional sphere. Therefore,*

$$\int_{\hat{\mathbf{R}}_i} p(\hat{\mathbf{R}}_i)\max(0,\cos\theta)^2 = \int_0^\pi \frac{S_{n-1}}{S_n}\cdot|\sin^{n-1}\theta|\cdot\max(0,\cos\theta)^2 \tag{10}$$

$$= \frac{S_{n-1}}{S_n}\int_0^{\pi/2}\sin^{n-1}\theta\cos^2\theta \tag{11}$$

$$= \frac{S_{n-1}}{S_n}\int_0^{\pi/2}\sin^{n-1}\theta(1-\sin^2\theta) \tag{12}$$

$$= \frac{S_{n-1}}{S_n}\int_0^{\pi/2}\sin^{n-1}\theta-\sin^{n+1}\theta \tag{13}$$

*Now we use a known result in existing literature that uses integration by parts to evaluate the integral of exponentiated sine function, which states,*

$$\int\sin^n\theta = -\frac{1}{n}\sin^{n-1}\theta\cos\theta + \frac{n-1}{n}\int\sin^{n-2}\theta \tag{14}$$

*Since our integration is between the limits $0$ and $\pi/2$, we find that the first term on the R.H.S. in the above expression is 0. Recursively expanding the $n-2^{th}$ power sine term, we can similarly eliminate all such terms until we are left with the integral of $\sin\theta$ or $\sin^0\theta$ depending on whether $n$ is odd or even. For the case when $n$ is odd, we get,*

$$\int_0^{\pi/2}\sin^n\theta = \left(\frac{2}{3}\cdot\frac{4}{5}\ldots\frac{n-1}{n}\right)\int_0^{\pi/2}\sin\theta \tag{15}$$

$$= -\left(\frac{2}{3}\cdot\frac{4}{5}\ldots\frac{n-1}{n}\right)\cos\theta\Big|_0^{\pi/2} \tag{16}$$

$$= \left(\frac{2}{3}\cdot\frac{4}{5}\ldots\frac{n-1}{n}\right) \tag{17}$$

*For the case when $n$ is even, we similarly get,*

$$\int_0^{\pi/2}\sin^n\theta = \left(\frac{1}{2}\cdot\frac{3}{4}\cdot\frac{5}{6}\ldots\frac{n-1}{n}\right)\int_0^{\pi/2}\sin^0\theta \tag{18}$$

$$= \left(\frac{1}{2}\cdot\frac{3}{4}\cdot\frac{5}{6}\ldots\frac{n-1}{n}\right)\int_0^{\pi/2}1 \tag{19}$$

$$= \left(\frac{1}{2}\cdot\frac{3}{4}\cdot\frac{5}{6}\ldots\frac{n-1}{n}\right)\cdot\frac{\pi}{2} \tag{20}$$

*Thus,*

$$\int_0^{\pi/2}\sin^{n-1}\theta-\sin^{n+1}\theta = \begin{cases} \frac{1}{n}\cdot\left(\frac{2}{3}\cdot\frac{4}{5}\ldots\frac{n-2}{n-1}\right) & \text{if } n \text{ is odd} \\ \frac{1}{n}\cdot\left(\frac{1}{2}\cdot\frac{3}{4}\ldots\frac{n-2}{n-1}\right)\cdot\frac{\pi}{2} & \text{otherwise} \end{cases} \tag{21}$$

*Define,*

$$K_n = \begin{cases} \frac{2S_{n-1}}{S_n}\cdot\left(\frac{2}{3}\cdot\frac{4}{5}\ldots\frac{n-2}{n-1}\right) & \text{if } n \text{ is odd} \\ \frac{2S_{n-1}}{S_n}\cdot\left(\frac{1}{2}\cdot\frac{3}{4}\ldots\frac{n-2}{n-1}\right)\cdot\frac{\pi}{2} & \text{otherwise} \end{cases} \tag{22}$$

*Then,*

$$\int_{\hat{\mathbf{R}}_i} p(\hat{\mathbf{R}}_i)\max(0,\cos\theta)^2 = \frac{0.5K_n}{n} \tag{23}$$

*Thus,*

$$\mathbb{E}[\|\mathbf{v}\|^2] = m\mathbb{E}[v_i^2] \tag{24}$$

$$= m \cdot \frac{2n}{m} \cdot \|\mathbf{u}\|^2 \cdot \frac{0.5K_n}{n} \tag{25}$$

$$= K_n \cdot \|\mathbf{u}\|^2 \tag{26}$$

*which proves the claim.* $\square$

**Lemma 1** *If network weights are sampled i.i.d. from a Gaussian distribution with mean 0 and biases are 0 at initialization, then conditioned on $\mathbf{h}^{l-1}$, each dimension of $\mathbb{1}(\mathbf{a}^l)$ follows an i.i.d. Bernoulli distribution with probability 0.5 at initialization.*

**Proof**: *Note that $\mathbf{a}^l := \mathbf{W}^l\mathbf{h}^{l-1}$ at initialization (biases are 0) and $\mathbf{W}^l$ are sampled i.i.d. from a random distribution with mean 0. Therefore, each dimension $\mathbf{a}_i^l$ is simply a weighted sum of i.i.d. zero mean Gaussian, which is also a 0 mean Gaussian random variable.*

*To prove the claim, note that the indicator operator applied on a random variable with 0 mean and symmetric distribution will have equal probability mass on both sides of 0, which is the same as a Bernoulli distributed random variable with probability 0.5. Finally, each dimension of $\mathbf{a}^l$ is i.i.d. simply because all the elements of $\mathbf{W}^l$ are sampled i.i.d., and hence each dimension of $\mathbf{a}^l$ is a weighted sum of a different set of i.i.d. random variables.* $\square$

**Theorem 2** *Let $\mathbf{v} = \sqrt{2} \cdot \left(\hat{\mathbf{R}}^T\mathbf{u}\right) \odot \mathbf{z}$, where $\mathbf{u} \in \mathbb{R}^m$, $\mathbf{R} \in \mathbb{R}^{m\times n}$ and $\mathbf{z} \in \mathbb{R}^n$. If each $\mathbf{R}_i \overset{i.i.d.}{\sim} P$ where $P$ is any isotropic distribution in $\mathbb{R}^n$ or alternatively $\hat{\mathbf{R}}$ is a randomly generated matrix with orthogonal rows and $\mathbf{z}_i \overset{i.i.d.}{\sim} \text{Bernoulli}(0.5)$, then for any fixed vector $\mathbf{u}$, $\mathbb{E}[\|\mathbf{v}\|^2] = \|\mathbf{u}\|^2$.*

**Proof**: *Our goal is to compute,*

$$\mathbb{E}[\|\mathbf{v}\|^2] = 2 \cdot \mathbb{E}[\|(\sum_{i=1}^{n} \hat{\mathbf{R}}_i u_i) \odot \mathbf{z}\|^2] \tag{27}$$

$$= 2 \cdot \mathbb{E}[\sum_{j=1}^{m}(\sum_{i=1}^{n} \hat{\mathbf{R}}_{ij} u_i)^2 \cdot z_j^2] \tag{28}$$

$$= 2 \cdot \mathbb{E}[z_j^2] \cdot \mathbb{E}[\sum_{j=1}^{m}(\sum_{i=1}^{n} \hat{\mathbf{R}}_{ij} u_i)^2] \tag{29}$$

$$= \mathbb{E}[\|(\sum_{i=1}^{n} \hat{\mathbf{R}}_i u_i)\|^2] \tag{30}$$

$$= \mathbb{E}[\sum_{i=1}^{n} u_i^2 \|\hat{\mathbf{R}}_i\|^2 + \sum_{i\neq j} u_i u_j \cdot \hat{\mathbf{R}}_i^T \hat{\mathbf{R}}_j] \tag{31}$$

$$= \|\mathbf{u}\|^2 + \sum_{i\neq j} u_i u_j \cdot \mathbb{E}[\hat{\mathbf{R}}_i^T \hat{\mathbf{R}}_j] \tag{32}$$

$$= \|\mathbf{u}\|^2 + \sum_{i\neq j} u_i u_j \cdot \mathbb{E}[\cos\phi] \tag{33}$$

*where $\phi$ is the angle between $\hat{\mathbf{R}}_i$ and $\hat{\mathbf{R}}_j$. For orthogonal matrix $\hat{\mathbf{R}}$ $\cos\phi$ is always 0, while for $\hat{\mathbf{R}}$ such that each $\mathbf{R}_i \overset{i.i.d.}{\sim} P$ where $P$ is any isotropic distribution, $\mathbb{E}[\cos\phi] = 0$. Thus for both cases[3] we have that,*

$$\mathbb{E}[\|\mathbf{v}\|^2] = \|\mathbf{u}\|^2 \tag{34}$$

*which proves the claim.* $\square$

**Theorem 3** *Let $\mathcal{R}(\{F_b(.)\}_{b=0}^{B-1}, \theta, \alpha)$ be a residual network with output $f_\theta(.)$. Assume that each residual block $F_b(.)$ ($\forall b$) is designed such that at initialization, $\|F_b(\mathbf{h})\| = \|\mathbf{h}\|$ for any input $\mathbf{h}$ to the residual block, and $< \mathbf{h}, F_b(\mathbf{h}) > \approx 0$. If we set $\alpha = 1/\sqrt{B}$, then,*

$$\|f_\theta(\mathbf{x})\|^2 \approx c \cdot \|\mathbf{x}\|^2 \tag{35}$$

*where $c \in [\sqrt{2}, \sqrt{e}]$.*

**Proof**: *Let $\mathbf{x}$ denote the input of the residual network. Consider the first hidden state $\mathbf{h}^1$ given by,*

$$\mathbf{h}^1 := \mathbf{x} + \alpha F_1(\mathbf{x}) \tag{36}$$

*Then the squared norm of $\mathbf{h}^1$ is given by,*

$$\|\mathbf{h}^1\|^2 = \|\mathbf{x} + \alpha F_1(\mathbf{x})\|^2 \tag{37}$$
$$= \|\mathbf{x}\|^2 + \alpha^2 \|F_1(\mathbf{x})\|^2 + 2\alpha < \mathbf{x}, F_1(\mathbf{x}) > \tag{38}$$

*Since $\|\mathbf{F}_1(\mathbf{x})\|^2 = \|\mathbf{x}\|^2$ and $< \mathbf{x}, F_1(\mathbf{x}) > \approx 0$ due to our assumptions, we have,*

$$\|\mathbf{h}^1\|^2 \approx \|\mathbf{x}\|^2 \cdot (1 + \alpha^2) \tag{39}$$

*Similarly,*

$$\mathbf{h}^2 := \mathbf{h}^1 + \alpha F_2(\mathbf{h}^1) \tag{40}$$

*Thus,*

$$\|\mathbf{h}^2\|^2 = \|\mathbf{h}^1\|^2 + \alpha^2 \|F_2(\mathbf{h}^1)\|^2 + 2\alpha < \mathbf{h}^1, F_2(\mathbf{h}^1) > \tag{41}$$

*Then due to our assumptions we get,*

$$\|\mathbf{h}^2\|^2 \approx \|\mathbf{h}^1\|^2 \cdot (1 + \alpha^2) \tag{42}$$

*Thus we get,*

$$\|\mathbf{h}^2\|^2 \approx \|\mathbf{x}\|^2 \cdot (1 + \alpha^2)^2 \tag{43}$$

*Extending such inequalities to the $B^{th}$ residual block, we get,*

$$\|\mathbf{h}^B\|^2 \approx \|\mathbf{x}\|^2 \cdot (1 + \alpha^2)^B \tag{44}$$

*Setting $\alpha = 1/\sqrt{B}$, we get,*

$$\|\mathbf{h}^B\|^2 \approx \|\mathbf{x}\|^2 \cdot \left(1 + \frac{1}{B}\right)^B \tag{45}$$

*Note that the factor $\left(1 + \frac{1}{B}\right)^B \to e$ as $B \to \infty$ due to the following well known result,*

$$\lim_{B \to \infty} \left(1 + \frac{1}{B}\right)^B = e \tag{46}$$

*Since $B \in \mathbb{Z}$, $\left(1 + \frac{1}{B}\right)^{B/2}$ lies in $[\sqrt{2}, \sqrt{e}]$.*

*Thus we have proved the claim.* $\square$

**Theorem 4** *Let $\mathcal{R}(\{F_b(.)\}_{b=0}^{B-1}, \theta, \alpha)$ be a residual network with output $f_\theta(.)$. Assume that each residual block $F_b(.)$ ($\forall b$) is designed such that at initialization, $\|\frac{\partial F_b(\mathbf{h}^b)}{\partial \mathbf{h}^b} \mathbf{u}\| = \|\mathbf{u}\|$ for any fixed input $\mathbf{u}$ of appropriate dimensions, and $< \frac{\partial \ell}{\partial \mathbf{h}^b}, \frac{\partial \mathbf{F}_{b-1}}{\partial \mathbf{h}^{b-1}} \cdot \frac{\partial \ell}{\partial \mathbf{h}_b} > \approx 0$. If $\alpha = \frac{1}{\sqrt{B}}$, then,*

$$\|\frac{\partial \ell}{\partial \mathbf{h}^1}\| \approx c \cdot \|\frac{\partial \ell}{\partial \mathbf{h}^B}\| \tag{47}$$

*where $c \in [\sqrt{2}, \sqrt{e}]$.*

**Proof**: *Recall,*

$$\mathbf{h}^b := \mathbf{x} + \alpha F_b(\mathbf{h}^{b-1}) \tag{48}$$

*Therefore, taking derivative on both sides,*

$$\frac{\partial \ell}{\partial \mathbf{h}^{b-1}} = (\mathbf{I} + \alpha \cdot \frac{\partial F_b}{\partial \mathbf{h}^{b-1}}) \cdot \frac{\partial \ell}{\partial \mathbf{h}^b} \tag{49}$$

$$= \frac{\partial \ell}{\partial \mathbf{h}^b} + \alpha \cdot \frac{\partial F_b}{\partial \mathbf{h}^{b-1}} \cdot \frac{\partial \ell}{\partial \mathbf{h}^b} \tag{50}$$

*Taking norm on both sides,*

$$\|\frac{\partial \ell}{\partial \mathbf{h}^{b-1}}\|^2 = \|\frac{\partial \ell}{\partial \mathbf{h}^b}\|^2 + \alpha^2 \cdot \|\frac{\partial F_b}{\partial \mathbf{h}^{b-1}} \cdot \frac{\partial \ell}{\partial \mathbf{h}^{b-1}}\|^2 + 2\alpha \cdot < \frac{\partial \ell}{\partial \mathbf{h}^b}, \frac{\partial F_b}{\partial \mathbf{h}^{b-1}} \frac{\partial \ell}{\partial \mathbf{h}^{b-1}} > \tag{51}$$

*Due to our assumptions, we have,*

$$\|\frac{\partial \ell}{\partial \mathbf{h}^{b-1}}\|^2 \approx \|\frac{\partial \ell}{\partial \mathbf{h}^b}\|^2 + \alpha^2 \cdot \|\frac{\partial \ell}{\partial \mathbf{h}^{b-1}}\|^2 \tag{52}$$

$$= (1 + \alpha^2) \cdot \|\frac{\partial \ell}{\partial \mathbf{h}^{b-1}}\|^2 \tag{53}$$

*Applying this result to all $B$ residual blocks we have that,*

$$\|\frac{\partial \ell}{\partial \mathbf{h}^1}\|^2 \approx (1 + \alpha^2)^B \cdot \|\frac{\partial \ell}{\partial \mathbf{h}^B}\|^2 \tag{54}$$

*Setting $\alpha = 1/\sqrt{B}$, we get,*

$$\|\frac{\partial \ell}{\partial \mathbf{h}^1}\|^2 \approx (1 + 1/B)^B \cdot \|\frac{\partial \ell}{\partial \mathbf{h}^B}\|^2 \tag{55}$$

*Note that the factor $\left(1 + \frac{1}{B}\right)^B \to e$ as $B \to \infty$ due to the following well known result,*

$$\lim_{B \to \infty} \left(1 + \frac{1}{B}\right)^B = e \tag{56}$$

*Since $B \in \mathbb{Z}$, $\left(1 + \frac{1}{B}\right)^{B/2}$ lies in $[\sqrt{2}, \sqrt{e}]$.*

*Thus we have proved the claim.* $\square$

## Footnotes

[2]Collecting 7M timesteps of experience took approximately 10h on a single GPU shared by 6 workers. Even though this amount of experience is enough to solve Pong, A3C usually needs many more interactions to learn competitive policies in more complex environments.

[3]This also suggests that orthogonal initialization is strictly better than Gaussian initialization since the result holds without the dependence on expectation in contrast to the Gaussian case.