[Reviews · NeurIPS 2019]

Reviewer 1



# Strengths - This submission copes with a problem of general importance: the optimization of weight-normalized networks. Coming-up with the right initialization strategy can prove to be critical in that case and the theoretical analysis suggests optimal init. strategy to avoid vanishing / exploding gradients in very deep networks. - The draft is clear, reads nicely and the research is well motivated. Allowing to improve the generalization capability of weight-normalized networks is an important research direction. - The proposed experiments seem to show the effectiveness of the initialization scheme. - The experiments reported in Table 1 and Table 2 are rather convincing. # Weaknesses / questions - First of all, I would like to know why is the assumption about asymptotic setting with infinite width needed. Where in the proof is it used? In terms of the parametrization proposed here it means that we work with n->infty. Eventually, in the experiments, the networks have rather small width compared to the depth. Following-up on that, could it be somehow empirically validated? Is this width assumption important? - What is the exact setup in the experiment presented in FIg. 2 (right). The text of the paper fails to exactly describe the experimental setup. What is the depth of the network considered here? - I find the experiments with resNets contestable. First of all the range of depths considered here is very surprising. Indeed, a resnet with 10000 blocks per stage is rather uncommon. Second, this experiment is ran for "one epoch of training" (l. 215) which in my opinion fails to show anything else than "PyTorch default init fails completely with such deep WN resnets". - The formulation on l. 243 is rather surprising: "Unlike previous works which use the test set for hyperparameter tuning". I do agree that proper hyper parameter tuning is essential in ML, but the wording is a bit hard in my opinion. Moreover, the "smaller training set" (l. 245) could be avoided by training the network with optimal parameters on the complete train+val. - In Table 1, I would expect a baseline WN model to be presented for other architectures than Resnet-110. Overall this is a good paper, coping with an interesting problem and is executed properly. I would like the authors to react to my negative comments / questions, and await the discussion period to make my final decision. # Rebuttal After reading the other reviews and the author's response, I decide to stick to my accept rating of 7. The author's response answered most of my concerns and doubts.

Reviewer 2



This paper proposed a new initialization strategy for weight normalized neural networks where the weight matrix in each layer is normalized. The key ingredient is to appropriately rescale the weight by a factor depending on the width. A theoretical analysis is provided showing that the proposed initialization strategy prevents vanishing/exploding gradients in expectation. Extensive experiments are provided showing that the proposed initialization does help the training and outperforms standard initialization strategy. Overall, I am very positive of the paper although I have several minor concerns, following are my comments. 1. It is mentioned several time in the paper that the initialization schemes is developed where the network width tends to infinity. I am a bit confused by it since none of the analysis explicitly used the infinite width condition. Maybe I am missing something, please clarify on it. 2. There is a closed form formula for the surface area of the unit ball, check for example Wikipedia, it would be better to prove that K_n=1 instead of a empirical check. 3. When performing the analysis on the backward pass, the gradients are evaluated according to intermediate pre-activation a^l. However, a^l is not a parameter that we are optimizing, what are the motivations of taking the derivatives with respect to a^l instead of W^l? 4 In the synthetic example in Figure 1, are the width of the fully connected network constant? In particular, the rescaling parameter in thm 1 and thm 2 match to each other when the width is constant, thus it is an extreme case. It would be good to perform experiments when the width varies and see if there is a difference. 5 If n_l > n_{l-1}, there is more rows than columns, how do we set the rows of the weight matrix to be orthogonal? 6 Is it possible to combine weight normalization with batch normalization? As far as I could see, batch normalization does not effect proposed strategy, maybe it would be good to try some experiments heuristically. ===Edit after Rebuttal=== I thank the authors for the clarification. I believe the initialization strategy introduced will be helpful for training deep networks in practice.

Reviewer 3



Summary ------- The paper proposes several initializations for weight normalized deep neural networks. The initializations are justified from simple theoretical considerations, then tested on numerous benchmarks, from which several conclusions are drawn. Originality and Significance --------------------------- While the justification of the initializations are simple and the approach is not new, the experiments provide a solid material for further exploration of weight normalized deep neural networks. In particular it seems to reduce the gap of performance of weight normalized networks compared to batch-normalized networks. This paves the way to better optimization of other structures such as in the reinforcement learning applications provided in the appendix. Quality and Clarity ------------------- The paper is well written and organized. In particular the experiments are well presented. - it is finally not clear what initialization is chosen for the residual networks, forward or backward ? - the authors could better emphasize that the plots separate the initialization of the scaling and the initialization of the weights. - also numerous plots are superimposed which makes their reading difficult. Conclusion: --------------- Overall the paper proposes a dense experimental study of initialization techniques for weight normalizations. This provides interesting material for future research. After rebuttal: ----------------- The authors answered my concern about the warm-up supplementary boosts and definitely showed the benefits of the approach. Overall this paper provides a very neat experimental material, the code is well written (I quickly skimmed through it) and therefore could be easily used for future research. The paper itself is well presented such that it can be used for the community. Overall I think the paper deserves publication. I hope that it could also open some discussions to relate weight normalization and the kernel perspective in the future.

[Author Response · NeurIPS 2019]

We thank reviewers for their valuable comments. We respond to the main concerns below.

**[R1/R2] Infinite width assumption:** the infinite width assumption is needed due to the technical detail that the norm
preservation of activations and gradients are studied in expectation over weights and not for a particular instance of
weights. Taking expectation is technically equivalent to considering infinite width and this strategy has been used in
previous papers that have studied weight initialization for un-normalized networks [5, 9].

**[R1] Setup in Fig. 2 (right):** we will clarify this in the revised manuscript. Both plots in Fig. 2 report results for the
same experiment, with the grid search parameters described in Table 1 in the appendix. The left plot shows the best
result at each depth, whereas the one on the right plots the accuracy for every job in our hyperparameter sweep.

**[R1] Very deep ResNet experiments:** Similar to that in Zhang et al. [31], we chose 10k block ResNet to stress the
point that our initialization indeed prevents gradient explosion/vanishing problem because otherwise we cannot train
at such depth. We run this experiment for 1 epoch for two reasons: (1) computational expense, and (2) to show that
training does not diverge at such depth when using our initialization with large learning rates compared to baselines.

**[R1] Train/test split (L243):** our intention was to highlight that the experimental setup is slightly different from that
in other works using the same architectures. We will rephrase L243 to better express this. We agree that training
on the complete train+val set with the best hyperparameters would boost performance, but we would also lose the
ability to track overfitting and perform early stopping. In order to reduce the impact of these factors and provide a fair
comparison, we decided to adopt the standard train/val/test split and follow best practices in hyperparameter tuning.

**[R2] Surface area of the unit ball:** while the closed form formula is available for high dimensional spheres, the ratio
seems hard to compute analytically because the constants and ratio of surface area do not trivially cancel out.

**[R2] Derivatives wrt $\mathbf{a}^l$:** it is sufficient to study the derivative with respect to pre-activations to show gradient
explosion/vanishing does not happen for weights. Derivative of weights depend on this term due to the chain rule.

**[R2] Layer width in Fig. 1:** width does indeed vary in Fig 1, as can be seen in the first line of `get_norm_ratios` in
the provided notebook (`synthetic_data_experiment.ipynb`). We will make this explicit in the revised manuscript.

**[R2] Orthogonal init when $\mathbf{n_l > n_{l-1}}$:** we followed the standard practice of orthogonal initialization, i.e. we
orthogonalize columns instead of rows as an approximation when $n_l > n_{l-1}$.

**[R3] Initialization scheme (forward/backward) for ResNets:** the initialization derived for forward and backward
pass for ResNet are identical. For fully connected layers within residual blocks, we use the initialization derived in
forward pass of fully connected layers (c.f. L180-183).

**[R3] Normalizing outputs of the ReLU as well:** an analysis of normalized ReLU output is beyond the scope of this
submission, as we focus on Weight Normalized networks (which only normalizes weights of the network). Therefore,
we believe that a comparison with the kernel counterparts of deep network would be distracting to the message of this
paper since our goal is specifically to study explosion/vanishing of activation and gradient in this network architecture.

**[R3] Complete definitions for ResNet:** everything is the same as in previous works except for the structure of residual
blocks (c.f. L176). This difference is described in L178 and illustrated in Figure 1 (right) in the supplementary material.

**[R1/R3] Additional WN baselines in Table 1:** we report two additional baseline results (proposed and data dependent
init without warmup) for each architecture and dataset in Table 1 below as suggested by the reviewers. Without warmup,
proposed init is better than data dependent init. Warmup improves the performance of proposed init (except on wide
ResNet, which we suspect happens due to large width where our theory holds more strongly).

| Dataset | Architecture | Method | Test Error (%) |
|---|---|---|---|
| CIFAR-10 | ResNet-56 | WN (proposed init + warmup) | $7.20 \pm 0.12$ |
| | | WN (proposed init + no warmup) | $7.87 \pm 0.14$ |
| | | WN (datadep init + no warmup) | $9.19 \pm 0.24$ |
| | ResNet-110 | WN (proposed init + warmup) | $6.69 \pm 0.11$ |
| | | WN (proposed init + no warmup) | $7.71 \pm 0.14$ |
| | | WN (datadep init + no warmup) | $9.33 \pm 0.10$ |
| | WRN-40-10 | WN (proposed init + warmup + cutout) | $4.75 \pm 0.08$ |
| | | WN (proposed init + no warmup + cutout) | $4.74 \pm 0.14$ |
| | | WN (datadep init + no warmup + cutout) | $6.10 \pm 0.23$ |
| CIFAR-100 | ResNet-164 | WN (proposed init + warmup + cutout) | $25.31 \pm 0.26$ |
| | | WN (proposed init + no warmup + cutout) | $27.30 \pm 0.49$ |
| | | WN (datadep init + no warmup + cutout) | $30.26 \pm 0.51$ |

[Meta-Review · NeurIPS 2019]

All reviewers are positive about the paper. The paper introduces a new initialization scheme for ResNets. The experimental results the authors present are extensive. The proposed initialization scheme appears to be quite effective empirically. The discussion of the results is particularly careful and nuanced. The contributions are of broad interest to the machine learning community. We recommend to take the reviewers' comments and suggestions into account while preparing the camera ready final version of the paper. Accept.